# Mission impossible accomplished? A European cross-national comparative study on the integration of the harm-benefit analysis into law and policy documents

Dominik Hajosi[1,2]*, Herwig Grimm[1]

1 Department of Interdisciplinary Life Sciences, Messerli Research Institute, University of Veterinary Medicine Vienna, Medical University of Vienna, University of Vienna, Vienna, Austria, 2 Institute of Comparative Medicine, Columbia University, New York, NY, United States of America

* 01145020@students.vetmeduni.ac.at

**Data Availability Statement:** All relevant data are within the paper and its Supporting Information files.

## Abstract

The harm-benefit analysis (HBA) is a cornerstone of the European Directive 2010/63/EU (the Directive). The Directive regulates the care and handling of animals used for scientific purposes in the European Union (EU). Since its implementation, there has been ongoing debate around the practical applicability of the HBA for research project review processes. The objectives of this study are to analyze the operationalization of HBA in EU member states and investigate the consistency of HBA's implementation in terms of national legislation and available policy documents. To meet these objectives, we evaluated the transposition of the HBA requirement into national legislation. We also conducted a comprehensive comparative cross-national analysis of all guidance documents pertinent to HBA. The results of our study show that there are (1) deficits in the transposition of the HBA requirement into national laws, (2) significant discrepancies in available policy documents relating to HBA, and (3) insufficiently consistent implementations of HBA in European countries.

## Introduction

In many countries, the use of animals for research purposes is strictly governed by designated laws. These laws ensure compliance with normative standards. The European Union's (EU) Directive 2010/63/EU (simply "the Directive" from here on) was published in 2010. It became a cornerstone legislative act regulating the use of animals for scientific purposes. EU member states were required to transpose the Directive's content into national laws by 2013 [1]. One of the Directive's purposes was to harmonize regulations and standards related to laboratory animals across EU member states.

Certain provisions within the Directive nonetheless allow EU member states some room to maneuver. The Directive's content can be interpreted by means of national contextualities. This naturally results in implementational differences when it comes to evaluating research projects using animals [2]. A report by the European Commission (EC) reviewing the

**Funding:** The author(s) received no specific funding for this work.

**Competing interests:** The authors have declared that no competing interests exist.

Directive's implementation over 5 years attests to this [3]. However, the Directive demands adherence to consistent standards for the evaluation of research projects (as described in Article 38). It states that projects should be assessed based on factors such as overall evaluation of objectives, severity of procedures, and compliance with the 3Rs (Replacement, Reduction, Refinement). A harm-benefit analysis (HBA) should also be an integral part of the process.

HBA can be considered a cornerstone for the evaluation of animal research projects. It is carried out by a competent authority, often with the support of review committees. The Directive (Art. 38 (2) d) describes HBA as part of a project evaluation designed to assess whether the harm to the animals in terms of suffering, pain, and distress is justified by the expected outcome taking into account ethical considerations, and may ultimately benefit human beings, animals, or the environment [1].

One finds similar scenarios in countries outside the EU. In the United States, the *Guide on the Care and Use of Laboratory Animals* (the *Guide*) serves as a primary reference document for programs using animals [4]. The *Guide* is also used as the basis for AAALAC (Association for Assessment and Accreditation of Laboratory Animal Care) International's accreditation process, which evaluates research programs globally. HBA has also been incorporated into various international guiding principles and multi-national agreements [5–7]. HBA is, then, crucial in the broader context of international research project evaluations.

Despite the above, some question the value of HBA as a practical tool in the assessment and decision-making process, especially since the Directive's issuance. Differences in the interpretation and implementation of HBA are central to the debate [7–11]. HBA is expected to facilitate decision-making processes around justifying the use of and harm inflicted on animals. It is supposed to do so by (1) defining a concept and (2) proposing a methodology to weigh harms against benefits. Herein lies a challenge. Multiple concepts and ideas have been described when it comes to practically dealing with HBA, especially as a legal requirement [7]. Ultimately, HBA should aid in avoiding unnecessary harm to animals, upholding the 3Rs and the welfare at stake, and ensuring transparent, objective, and reasonable justifications for the intended scientific use of animals.

Given the above, we maintain that it is crucial for the international research community to apply consistent standards. This can help achieve regulatory harmonization and avoid moral dumping (i.e., attracting research through lowering standards).

Our primary aim in this study is to investigate similarities and differences related to the implementation of HBA in European countries, particularly EU member states. However, the Directive is not explicit about implementation details for carrying out HBA. As such, we hypothesize that there are different approaches available to operationalizing HBA on the national level. HBA has been inconsistently implemented in European countries.

The EC has established various expert working groups. These groups have provided guidance documents to accompany the Directive [12–16]. One such document relates to project evaluation and HBA. The document is supposed to be used on a supranational level to facilitate HBA during project evaluation processes. The problem is that these guidance documents are not legally binding (even if they are often accepted). It remains unclear how HBA should be practically implemented during project evaluations [11, 17–21].

We shall investigate deficits in HBA's harmonization across Europe. We shall also explore potentially promising ways to move forward based on national laws and policy documents. As such, our research question can be concisely stated as follows:

*To what extent has HBA been consistently implemented and operationalized in European countries' legislations and policy documents?*

## Materials and methods

### Scope and strategy of the study

Regarding geographical scope, our study mostly focuses on EU member states. We are, though, also interested in non-EU member European countries like the United Kingdom (UK), Norway (NO), and Switzerland (CH). This is for the following reasons: the UK was an EU member state at the time when the Directive was supposed to be implemented into EU member states' national laws; NO has been implementing EU standards in animal research and, since 2018, the country has also been included in EU statistics on the use of animals for scientific purposes [22, 23]; CH's legislation on animal research resembles the Directive in significant respects [24].

We believe that including the UK, NO, and CH in our study strengthens the relevance of our research. We will, however, treat them separately in our analysis and results because they are not EU member states.

We applied a three-step strategy to answer our research question: (1) we analyzed national laws and applicable legal documents to show the extent of the HBA requirement's transposition; (2) we performed a search to identify relevant guidance documents. We did so by carrying out a comprehensive literature survey using online databases and search engines. We also directly contacted the relevant national committees of EU member states; (3) we performed a comparative cross-national analysis of the available guidance documents and HBA's implementation.

We now describe these three steps in detail.

### Transposition of HBA into EU member states' laws and regulations

The Directive has been made available in 23 national languages of EU member states [1]. We based our search for the transposition of the Directive's HBA requirement on an analysis of the Directive's various translations. We compared them directly with the national laws of the relevant countries. We paid special attention to whether the Directive's wording had been changed and/or whether modifications had been made to formulate national laws. We also searched for additional published legal bylaws, references, or footnotes having legal character and relevance to HBA and our research question. We used translation software (Google Translate, Microsoft Translate, or DeepL) when an English translation of national laws was unavailable. Our goal in this method was to elucidate consistencies and discrepancies in the transposition of the Directive's HBA requirement by addressing questions asked in Table 1.

### Availability of guidance documents

As a second method, we performed a comprehensive search for guidance documents addressing HBA's practical implementation in project evaluation processes. We have included all

**Table 1. Research questions: Transposing the Directive's HBA requirement.**

| | Transposition of HBA requirement into national laws | Answer options |
|---|---|---|
| (a) | Are there differences in the wording of the Directive's HBA requirement in the various EU member states' national laws? | Yes/No |
| (b) | If "Yes", then what differences are there in the wording of the EU member states' national laws compared with the Directive? | If the answer to (a) is "Yes", provide additional information regarding the differences |
| (c) | Does the EU member state have any supplementary legal bylaws specifically referring to HBA? | Yes/No |

**Table 2. Research questions: Availability of guidance documents in European countries.**

|  | Availability of guidance documents | Answer options |
|---|---|---|
| (a) | Is a national guidance document available for the respective country? | Yes/No |
| (b) | Is a regional guidance document available for the respective country? | Yes/No |
| (c) | Did the national committee respond to the e-mail inquiry? | Yes/No |

guidelines, guidance documents, policies, and recommendations that we found in a literature search and that were made available to us by national committees. We distinguished between national and regional guidance documents because there are significant differences between EU member states' project evaluation processes [2]. Regional guidance documents can fall outside the national scope of any given EU member state. We performed the search through a systematic literature survey and by contacting respective national committees via e-mail to inquire about available guidance documents. Table 2 summarizes the research questions for this part of the study.

## Literature search terms and streams

We selected our search terms based on whether we thought that those terms would be included in the published research articles. The search terms are summarized in Fig 1.

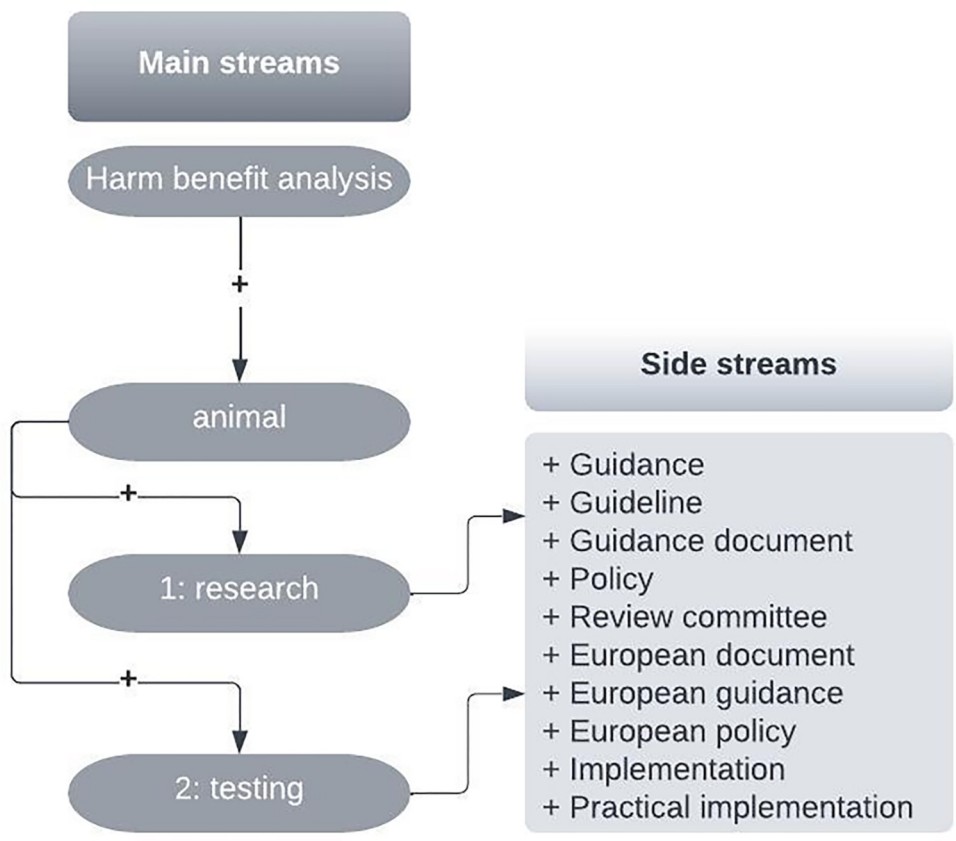

**Fig 1. Schematic methodology for creating all search streams with an overview of search terms used.** We performed our initial search by using the terms "harm-benefit analysis", "animal", "research", and "testing". Our selected combinations of these terms created two separate streams: 1. Main Stream 1: "harm benefit analysis + animal + research". 2. Main Stream 2: "harm benefit analysis + animal + testing".

We then differentiated these two main streams into side streams by adding several additional terms. Fig 1 shows how we established the main streams and side streams, and which terms we employed in the survey. We then used each stream's results (for the respective database) to perform further analyses and article selections.

We also reviewed all the selected literature's listed references and employed the "snowballing method" to identify additional material. We assessed all references based on our inclusion and exclusion criteria. References that we judged to be relevant were included in this study and further analyzed.

The created main + side streams and the "snowballing method" were also used for finding literature using the search engines Google and Norecopa.

## Database selection

We selected the following databases for our literature search: PubMed, Scopus, ISI Web of Science, and Ingenta Connect. Our choice was based on the fact that all four databases provide comprehensive coverage relevant to laboratory animal science, animal research ethics, biomedical research, and the life sciences in general. We also included the search engines Google and Norecopa [25] in our search strategy. Norecopa is a national consensus platform of Norway and provides a well-structured and up-to-date database with significant information relevant to the international animal research community.

## Article selection and inclusion/exclusion criteria

We searched for relevant scientific articles, book chapters, and publicly available legal or policy documents according to the criteria in Table 3.

## Data analysis and extraction

We carefully analyzed each article, book chapter, and legal or policy document. We began by screening the title, abstract, and introduction for relevance to our study. We also checked whether they met our inclusion/exclusion criteria. Where necessary, we then performed a full-text analysis. We performed full-text analyses on all documents addressing operationalization and/or guidance on HBA's practical implementation in project evaluation processes. We extracted relevant data from the full-text analyses by examining information directly related to our research question.

## Directly contacting national committees

We identified the contact details of national committees for all countries forming part of our study. We then sent e-mails inquiring whether a guidance document for practically

**Table 3. Inclusion and exclusion criteria used for article selection.**

| | |
|---|---|
| **Inclusion criteria** | 1) **Language:** English or other relevant national language.<br>2) **Timeline**: Publication of the Directive (September 22, 2010) up until July 31, 2023.<br>3) **Scope:** All EU member states plus the UK, CH, and NO.<br>4) **Content:** HBA pertaining to the use of animals in research. |
| **Exclusion criteria** | 1) **Language:** Not English or other relevant national language.<br>2) **Timeline:** Any material published before September 22, 2010 or after July 31, 2023.<br>3) **Scope:** Any country that is not an EU member state, the UK, CH, and NO.<br>4) **Content:** HBA not pertaining to the use of animals in research. |

National documents that were unavailable in English were translated by native speakers or by translation software (Google Translate, Microsoft Translate, and/or DeepL). Where possible, native speakers cross-checked the software translations for accuracy.

implementing HBA was available and being used by review committees. We sent follow-up e-mails to national committee contacts who did not respond to the initial request or when we needed further clarifications. We considered all responses received up until August 31, 2023.

## Comparative, cross-national analysis of HBA implementation

As a final step in our study, we performed a comparative and cross-national analysis of all available guidance documents' interpretations and practical implementations of the Directive's HBA requirement. To do so, we divided the original formulation of the Directive's HBA requirement into its five elements, which will be addressed as *domains* in the following: (1) *harm to animals*, (2) *justification of the harm*, (3) *outcome*, (4) *ethical considerations*, and (5) *benefits to human beings, animals, or the environment*.

We also formulated two questions for the five domains. Our analysis is based on the answers to those questions (Table 4). We expected to achieve a consistent comparison of HBA implementations by using questions that were predefined and standardized for all domains.

## Results

### Transposition of the HBA requirement into EU member states' laws

Our results show that there are differences in the wording of the Directive's HBA requirement and its transposition into some EU member states' laws (Fig 2).

We noted discrepancies between the Directive's formulation and the national laws of France, Germany, Finland, Denmark and Sweden.

Both the French and German legislations describe the HBA requirement's domains. There is, however, a difference in how those domains are integrated into the legislation's relevant paragraphs. The German legislation modifies the Directive's wording so that the domains are listed as a series of criteria that must be fulfilled during project evaluations [27]. The French legislation excludes the phrase "taking into account ethical considerations" from the

**Table 4. HBA domains and overview of questions used for analysis.**

| HBA domains formulated in the Directive | | Questions to systematize analysis of guidance documents | Answer options |
|---|---|---|---|
| **(1)** | ***Harm to animals*** | Is the domain clarified/defined in the guidance document? | Yes/No |
| | | Does the guidance document provide a method for practically including the domain in HBA? | Yes/No |
| **(2)** | ***Justification for the harm*** | Is the domain clarified/defined in the guidance document? | Yes/No |
| | | Does the guidance document provide a method for practically including the domain in HBA? | Yes/No |
| **(3)** | ***Outcome*** | Is the domain clarified/defined in the guidance document? | Yes/No |
| | | Does the guidance document provide a method for practically including the domain in HBA? | Yes/No |
| **(4)** | ***Ethical considerations*** | Is the domain clarified/defined in the guidance document? | Yes/No |
| | | Does the guidance document provide a method for practically including the domain in HBA? | Yes/No |
| **(5)** | ***Benefits to human beings, animals, or the environment*** | Is the domain clarified/defined in the guidance document? | Yes/No |
| | | Does the guidance document provide a method for practically including the domain in HBA? | Yes/No |

| | | EU Member States | | | | | | | | | | | | | | | | | | | | | | | | | |
|---|---|---|---|---|---|---|---|---|---|---|---|---|---|---|---|---|---|---|---|---|---|---|---|---|---|---|---|
| | | AT | BE | BG | HR | CY | CZ | DK | EE | FI | FR | DE | GR | HU | IE | IT | LV | LT | LU | MT | NL | PL | PT | RO | SK | SI | ES | SE |
| Research Questions | Are there differences in the wording of the Directive's HBA requirement in the various EU member states' national laws? | •¹ | — | — | — | — | — | • | — | • | • | • | — | — | — | — | — | — | — | — | — | — | — | — | — | — | — | • |
| | Does the EU member state have any supplementary legal bylaws specifically referring to HBA? | •² | — | — | — | — | — | — | — | — | — | — | — | — | — | — | — | — | — | — | — | — | — | — | — | — | — | — |

**Fig 2. Overview of the HBA requirement's transposition into EU member states' national legislations.** "•" (Yes = differences were identified) or "—" (No = no differences were identified). ¹ addition of criteria catalog; ² regulation regarding the criteria catalog (TVKKV [26]).

Directive's wording. An *ethical* aspect is, nonetheless, implied when the term "project evaluation" is rephrased as "ethical evaluation": "The ethical evaluation of projects is carried out at a level of detail appropriate to the type of project [. . .]" [28] [Authors' note: Translation DeepL].

Finnish law uses notably different terminology. It states that project evaluation is granted if, among other things, "the expected benefit from the project to human beings, animals or the environment is in an ethically justified proportion considering the harm to the animals" [29].

The Danish legislation also uses wording that differs from the Directive. It implies that a negative HBA can result in a project proposal's rejection: "The Animal Experiments Inspectorate may refuse to grant permission for animal experiments if the experiment is not deemed to be of significant benefit, including if the stress to which the animal is exposed is not commensurate with the usefulness of the experiment and the product" [30] [Authors' note: Translation DeepL].

The transposition in Sweden's legislation contains a terminological difference. The Directive's term "harm" is replaced with "suffering" and it is stated that the "[. . .] suffering of the experimental animal shall be weighed against the expected benefit [. . .]" [31] [Authors' note: Translation DeepL]. Even though there is an apparent difference, the term "suffering" itself amounts to the key element of the Directive's pathocentric harm domain. Given that this is a minor difference in the transposition, we are describing it in this section, but we do not discuss it in more detail further on.

Austria's national legislation includes an addition to the HBA requirement. It states that the so-called "criteria catalog" should be considered part of HBA. The goal is to objectify HBA by establishing specific criteria that must be met [26]. The criteria catalog mentions HBA's predefined five domains and it will be addressed further in the comparative analysis of this study.

## Availability of guidance documents

**Results of the literature survey.** We have summarized our results for the respective databases' search streams in Figs 3 and 4.

We examined 199 articles. We selected literature that was relevant to the overall research field and questions. However, no articles were classified as guidance documents related to HBA's operationalization.

Our literature search yielded six documents (and one exception, discussed in the following section) that could appropriately be categorized as guidance documents. We, therefore, included them in our study for further consideration.

**Responses from national committees.** We contacted national committees (*n* = 30) from all countries included in our study to inquire about the existence of a guidance document. We received feedback from 22 committees (Fig 5).

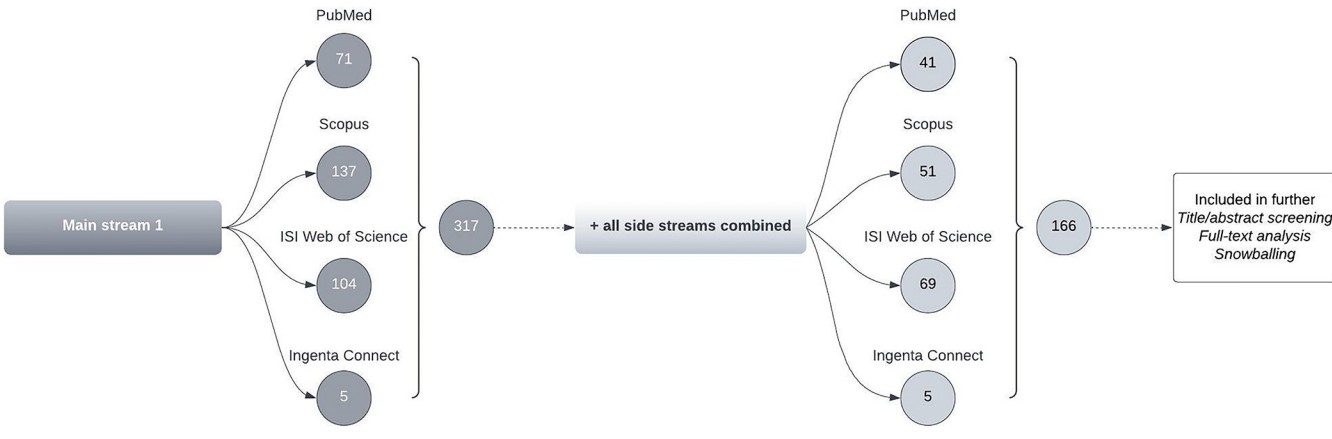

**Fig 3. Overview of all search stream results for Main Stream 1.**

Out of 22 committees, the Belgian, Dutch, and UK national committees reported that a national or regional guidance document pertaining to HBA's operationalization was available. We also found these documents through our literature survey. The NO national committee reported that there was no guidance document available. Our literature survey, nonetheless, identified a Norwegian document addressing HBA. However, we did not deem this document to be appropriate for further analysis (we address this case further in the following section). As such, we answered "No" to the question of whether a national and/or regional guidance document is available in NO.

## National and regional guidance documents

Our literature survey results show that national-level guidance documents are available in four countries: France, the Netherlands, Switzerland, and the UK. Belgium (BE) has published a regional guidance document that is relevant only to the Brussels-Capital region. We also included Austria's legally required criteria catalog as a type of guidance document. The European working group's guideline on project evaluation and retrospective assessment (HBA-EU) serves as an overarching document on the supranational level. We included it in our analysis

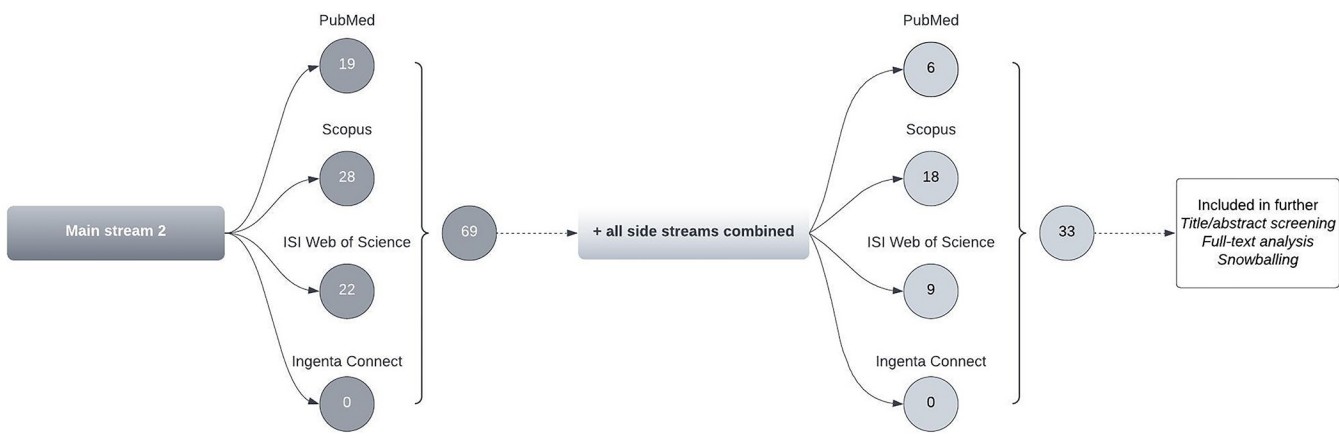

**Fig 4. Overview of all search stream results for Main Stream 2.**

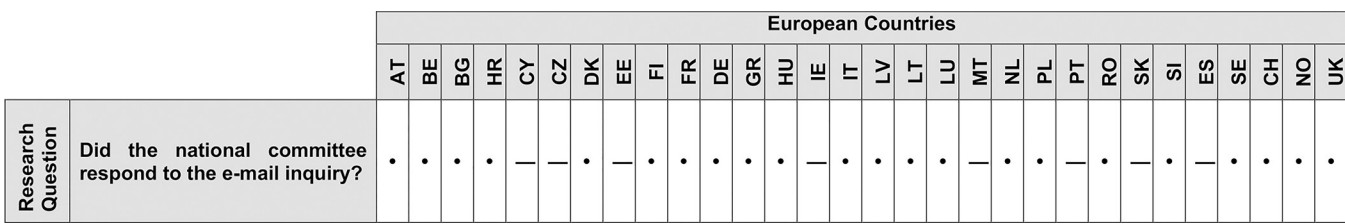

| | | European Countries | | | | | | | | | | | | | | | | | | | | | | | | | | | | | |
|---|---|---|---|---|---|---|---|---|---|---|---|---|---|---|---|---|---|---|---|---|---|---|---|---|---|---|---|---|---|---|---|---|
| | | AT | BE | BG | HR | CY | CZ | DK | EE | FI | FR | DE | GR | HU | IE | IT | LV | LT | LU | MT | NL | PL | PT | RO | SK | SI | ES | SE | CH | NO | UK |
| Research Question | Did the national committee respond to the e-mail inquiry? | • | • | • | • | — | — | • | — | • | • | • | • | • | — | • | • | • | • | — | • | • | — | • | — | • | — | • | • | • | • |

**Fig 5. Responses from contacted national committees by country.** "•" (Yes) or "—" (No).

because it can be regarded as a primary reference for EU member states' reviewing bodies when it comes to project evaluations (Fig 6).

The NO national committee has published a document called "Ethical Guidelines for the Use of Animals in Research". It contains basic principles relevant to the use of animals in research. One of these principles relates to the proportion of suffering versus benefit [32]. That said, the key message remains somewhat superficial. We excluded this document from further analysis because it did not appear to be relevant to our research or our analysis. NO's national committee also did not provide us with the document, stating that no guidance documents were available, which substantiated our decision.

Animal testing in Austria is regulated by the Animal Testing Act [33]. The act is complemented by regulations such as the Regulation on the Criteria Catalog for Animal Experiments (HBA-AT) [26]. We included this regulation as a guidance document because (1) it enforces the Animal Testing Act and (2) it establishes predefined criteria for an objective HBA of procedures. HBA-AT is a legal document formulated by the Austrian Federal Ministry of Science, Research, and Economy. Applicants are, therefore, required to complete it. It can be accessed through Austria's online legal information system.

The Brussels Commission for Animal Experimentation and the Belgian authorities (*Bruxelles Environnement*) have developed a guidance document (HBA-BE) for practical implementations of HBA during research project evaluations. This document is intended to facilitate HBA processes performed by an ethical committee in the Brussels-Capital Region. It can be found on the Brussels Environment website [34].

In the Netherlands (NL), the Central Committee on Animal Experiments (*Centrale Commissie Dierproven*, CCD) has produced a similar guidance document (HBA-NL). The CCD authorizes project applications and seeks advice from the Animal Experiments Committee (Dierexperimentencommissie, DEC), which plays an advisory role during project evaluation (including HBA). The DEC also gives a recommendation to the CCD related to project authorizations. The CCD's document is published on the CCD website [35].

The French guidance document (HBA-FR) in our study was published by Gircor, a national think tank and non-profit animal research advocacy organization aiming to facilitate

| | | European Countries | | | | | | | | | | | | | | | | | | | | | | | | | | | | | |
|---|---|---|---|---|---|---|---|---|---|---|---|---|---|---|---|---|---|---|---|---|---|---|---|---|---|---|---|---|---|---|---|---|
| | | AT | BE | BG | HR | CY | CZ | DK | EE | FI | FR | DE | GR | HU | IE | IT | LV | LT | LU | MT | NL | PL | PT | RO | SK | SI | ES | SE | CH | NO | UK |
| Research Questions | Is there a national guidance document available in the respective country? | • | — | — | — | — | — | — | — | — | • | — | — | — | — | — | — | — | — | — | • | — | — | — | — | — | — | — | — | • | — | • |
| | Is there a regional guidance document available in the respective country? | — | • | — | — | — | — | — | — | — | — | — | — | — | — | — | — | — | — | — | — | — | — | — | — | — | — | — | — | — | — | — |

**Fig 6. Available guidance documents sorted by country.** "•" (Yes) or "—" (No).

stakeholder dialogues related to animal research. The HBA-FR is intended to guide review committees' project evaluation processes. The document can be found online [36].

The Swiss guideline (HBA-CH) was developed by the Ethics Committee for Animal Experimentation. The committee has advisory responsibilities in the field of animal experimentation and is co-directed by the Swiss Academy of Medical Sciences and the Swiss Academy of Sciences. The Swiss guideline is available on the Swiss Academies of Arts and Sciences' website [37]. The Swiss Federal Food and Safety Veterinary Office (FSVO) has also developed a guide proposing a uniform, step-by-step approach to weighing stakeholder interests [38, 39].

The UK guidance document (HBA-UK) by the Animals in Science Regulation Unit, which is part of the UK Home Office, has been identified and included in this study as well. The Animals in Science Regulation Unit Inspectorate performs HBA as part of the research project authorization process. The UK document is published by the Home Office and can be found online [40].

## Comparative, cross-national analysis of HBA implementation

Based on the pre-defined domains and the available guidance documents, we performed a comparative and cross-national analysis of HBA implementations. We have highlighted the results for the EU (as the overarching framework), the UK, and CH (as both are non-EU members) in different colors in all relevant tables.

We now analyze the domains in more depth, specifically their manifestation in the guidance documents.

## Domain: Harm to animals

We searched each document for a definition of *harm*. The Directive defines *harm* in Art. 38 (2) d) as *"suffering, pain, and distress"* inflicted on animals [1]. It does not provide a more detailed definition, leaving further classifications open. However, Annex VIII in the Directive lays down provisions related to a variable classification of a procedure's severity. The severity classification is carried out by assessing the degree of overall *harm* experienced by the animal most affected during a procedure [1]. A summary of the analysis of the harm domain can be found in Table 5.

HBA-EU refers to *harm* as something that must be assessed and accounted for in HBA. The document focuses on the severity classification for procedures as a tool for the practical implementation of this domain. The severity classification of a procedure follows four categories: non-recovery, mild, moderate, and severe. Alongside secondary factors, the most suitable severity category is judged according to the most severe effects an individual animal is likely to experience [1], but it may not reflect the overall welfare costs: "It is important therefore to know what the predicted severity is for *all animals* used on the procedure, taking into account the methods used to minimize adverse effects." [16].

**Table 5.** *Harm* domain in European HBA guidance documents.

| | | | European countries | | | | | | |
|---|---|---|---|---|---|---|---|---|---|
| | | | EU | AT | BE | FR | NL | CH | UK |
| *Domain* | **Harm to animals** | Q1 | • | • | • | • | • | • | • |
| | | Q2 | • | • | • | • | • | • | • |

Q1: "Is the domain clarified/defined in the guidance document?".

Q2: "Does the guidance document provide a method for practically including the domain in HBA?". Answer options for Q1 and Q2 were either "•" (Yes) or "–" (No).

In Austria (AT), HBA-AT asks applicants to declare the total number of animals that will be used in a study. It also requires that animals be allocated to the respective severity categories. This reflects a differentiated view on the overall anticipated harm inflicted on all animals. HBA-AT also requires applicants to justify the expected harm, which may include ethical considerations. This requires specifying the total number of animals assigned to severity categories and a justification for the numbers allocated to the different categories [26].

HBA-BE describes harm as adverse welfare effects experienced by animals during a project. Harm can affect animals immediately or be delayed. It can involve direct harm, e.g., during a procedure, denying animals certain pleasures or suppressing certain enrichments. Animals can also be aware or unaware of the harm being inflicted. In general, one can, thus, distinguish between project-related harm and contingent harm. The Belgian document refers to "Five Freedoms" developed by the Farm Animal Welfare Council (FAWC) in light of the Brambell Report [41]. This framework has been adapted to farm animals and animals used for scientific purposes [42, 43] and it is used in the HBA-BE to evaluate negative welfare effects originating in the following areas (1) nutrition, (2) environment, (3) health, (4) behavior, and (5) mental state/experience. A project proposal must also consider modulating factors:

> Each modulating factor for harm may mitigate or aggravate the harm inflicted on the animals due to the experiment. The effect may only be aggravating or mitigating, but both effects are possible as well [34].

Modulating factors represent some of the key issues described in the HBA-EU document. They consist of the following components: (1) a description of methods used to control adverse effects, (2) frequency and duration of procedures, (3) duration in proportion to the animal's lifespan, (4) severity level assessment, (5) the animal's species, (6) total number of animals, (7) a description of the experiment's termination and humane endpoints, (8) the animal's overall life experience, (9) the animal's health status, (10) housing details, (11) care provided, (12) monitoring regime, (13) genetic modulations and their impact, (14) the competence of animal care and research personnel, (15) the animal's origin, and (16) transportation [16, 34].

HBA-FR states that a researcher must identify harm in the scientific project proposal and also address and define the overall harm that animals are likely to experience. This must be done in reference to a severity classification and it must include all adverse physical and psychological effects [36].

HBA-NL defines harm as distress caused either by direct, invasive means (e.g., during experiments) or indirect, non-invasive means (e.g., housing conditions or transportation). Harm can also result from cumulative effects. It must also be classified as either non-recovery, mild, moderate, or severe. This classification is based on both the degree of pain and suffering and the lasting harm an animal experiences and it concords with the Directive [35]. Violations of the integrity (or the wholeness) of an animal must also be assessed. An animal's integrity can be influenced by physical, behavioral, or psychological factors.

HBA-CH defines harm according to a legal definition in the Swiss Animal Protection Act [44]. Harm is divided into pathocentric harm and non-pathocentric harm.

> Pathocentric harm: Animals are subjected to pain, suffering, injury, distress, or degradation. The animal has a negative subjective experience; it experiences the harm as burdensome.

> Non-pathocentric harm: Animals are subjected to degradation, excessive instrumentalization, or significant interventions in their appearance or capacities. The animal does not

have necessarily a negative subjective experience; it does not automatically experience the harm as burdensome [37].

Both kinds of harm occur in animal experiments, and both must be considered during HBA. A severity degree (0–3)–that is different to the Directive's classification–is assigned to experiments causing pathocentric harm. This is taken into account during an HBA weighing process. Non-pathocentric harms are not classified on a scale because the animal does not experience such harms as burdensome. However, they might affect the *dignity* of an animal in a certain way [37]. The dignity concept–defined as the inherent value of the animal–is unique to the Swiss Animal Protection Act [37, 44]. The inherent value needs to be respected by anyone who is handling it, and it is affected or not respected if an animal is subjected to harms that are not justified by outweighing interests [37].

HBA-UK describes the severity classification as a way to provide information about harms anticipated during a project. In general, severity is classified as either non-recovery, mild, moderate, or severe. The document states, however, that this classification does not adequately reflect anticipated harm. Additional factors must also be considered. Some of these include (1) contingent harms, (2) project-related harms, (3) duration or incidence of harm, and (4) cumulative effects [40]. A numerical scoring of harm is not used, but the four severity categories may allow for a differentiation of harm for the project evaluation. Hence, they follow the HBA-EU in this regard.

## Domain: Justification of harm

The Directive stipulates a justification requirement as an element of the HBA. We, therefore, included the justificatory domain in our study. Art. 38 (2) of the Directive states as follows: "The project evaluation shall consist of [. . .] a harm-benefit-analysis, to assess whether the harm to the animals [. . .] is justified by the expected outcome [. . .]" [1]. This requires a defined justificatory relationship between harm and benefit. We searched all available material for practical guidance on operationalizing this justificatory domain. The results can be found in Table 6.

HBA-EU mentions the need for justifications for several aspects of a project application (e.g., animal use and expected harm). However, a detailed methodology for how to carry out the required justification was not available. HBA-EU provides examples of methods for achieving a justification but does not place them in a practical context. The document does, nonetheless, suggest that the Bateson Cube could facilitate project assessments [16]. It also proposes a modified version of the Bateson Cube with a color-coding scheme to assess harms and benefits. HBA-EU defines the justificatory domain in a formula (Fig 7).

HBA-EU also states that no single approach is satisfactory:

**Table 6.** *Justification of harm* **domain in European HBA guidance documents.**

| | | | European countries | | | | | | |
|---|---|---|---|---|---|---|---|---|---|
| | | | **EU** | **AT** | **BE** | **FR** | **NL** | **CH** | **UK** |
| *Domain* | **Justification of harm** | **Q1** | • | • | – | • | • | • | • |
| | | **Q2** | (•) | – | – | – | – | • | – |

Q1: "Is the domain clarified/defined in the guidance document?".

Q2: "Does the guidance document provide a method for practically including the domain in HBA?". Answer options for Q1 and Q2 were either "•" (Yes) or "–" (No). Only one answer was inconclusive; it is represented as "(•)".

$$\text{Justification} = \frac{\text{Importance of objectives} \times \text{Probability of achievement}}{\text{Harms to animals}}$$

**Fig 7. Definition of justification in HBA-EU [16].**

The evaluation process is multi-factorial, and no simple numerical allocation formula can provide a simple yes/no answer. Knowledge of the different published models of harm/benefit analysis is needed. These systems can be useful tools for discussion to ensure all issues are given structure and systematic consideration but these should not be used in isolation to replace intelligent interpretation of the information provided [16].

As such, HBA-EU remains vague when it comes to operationalization. It merely describes potential approaches.

HBA-AT requires applicants to explain the harm inflicted on animals. One can consider such an explanation to be a justification. However, HBA-AT does not provide any additional, transparent recommendations on how to operationalize the justificatory component (e.g., a methodology to weigh identified harms and benefits).

HBA-BE recognizes that weighing harms against benefits is not a simple and uniform process: "A unique, case-by-case evaluation for each proposed project in which the importance and magnitude of the benefit is assessed will have to be performed by an ethical commission" [34]. HBA-BE suggests a color-coded scheme to identify criteria that should be taken into consideration during project assessment. These criteria include primary benefits, the likelihood of achieving benefits, main harms, and modulating harm factors. However, the color-coding of the criteria merely identifies them, and the document does not describe an implementation methodology nor provide a detailed weighing strategy.

HBA-FR states that HBA must justify harm inflicted on animals against the project's expected results by applying a certain value to all information available in the project application. Since the "certain value" is not further specified, it becomes the reviewing person's responsibility.

HBA-NL describes a "central moral question" [Authors' note: Translation DeepL], one directly related to the *value* of the project. This question should clearly articulate the problem at stake, and must be answered with "Yes" or "No". A "Yes" means that the harm is justified; the outcome is highly valued. Harm and outcome/benefits to relevant stakeholders can be illustrated in a matrix (Table 7). Stakeholders include animals involved in the research, researchers, human beings, and other entities potentially relevant to the project (e.g., the environment and general society).

One answers the moral question by justifying or weighing the moral values at stake. The reviewing committee is expected to assign a value, and these values are then weighed against each other. A matrix can provide a visual correlation between core values affecting individual stakeholders and anticipated benefits. However, it does not attribute weights to individual values or practically describe interpreting the matrix. In sum, HBA-NL remains unclear when it comes to a clear method for carrying out a justification; it remains vague on the practical level.

The HBA-CH is again an outlier. It requires a researcher to explain the proposed project's essentiality by assessing instrumental essentiality and goal-related essentiality. Instrumental essentiality applies if (1) the proposed experiments are reproducible, generalizable, and sufficiently robust to achieve the expected results, (2) the experiments are necessary to achieve the

**Table 7. Matrix in HBA-NL [35].**

| *Moral values* Stakeholders | *Welfare* | *Autonomy* | *Justice* |
|---|---|---|---|
| **Test animals** | Health | Natural behavior | Alternatives |
| | Pain | | Intrinsic value |
| | Stress | | Integrity |
| **Target animals** | Health | Natural behavior | |
| | Pain | | Proportionality |
| | Stress | | Intrinsic value |
| **Target group(s) project** | Quality | Choice | |
| | Safety | | Availability of, e.g., the product tested |
| | Health | | Proportionality |
| | Pain | | |

expected results, and (3) the experiments cannot be performed without the use of animals. A justification is made via an assessment of goal-related essentiality. The assessment can show that the expected goal of a proposed experiment outweighs the expected harm to animals. A justification is met when the interests of society are suitably weighed against harms inflicted on animals. A document published by the Federal Food Safety and Veterinary Offices (FSVO) provides a matrix for reading the results (Table 8) [39]. Legitimate interests can gain a weight of 4 (****), while strains can gain a maximum weight of 3 (***). The dignity of an animal is respected if the weight of legitimate interests outweighs the weight of harm.

HBA-UK describes the weighing of harms against benefits as a process leading to the conclusion that overall anticipated harm is justified by expected benefits. The information provided in the project proposal should help the reviewing authority to determine an "overall judgment of the final 'weight' (i.e., extent, degree, severity) of the harms [. . .]" [40]. However, setting a value for benefits is more challenging than for harms because it is difficult to predict a factor of uncertainty. To determine the value of benefits, the authority must consider the importance of benefits and the likelihood of achieving them. A lower or higher value can then be assigned to the benefits. The decision is a "value-laden judgment" that is influenced by a variety of factors [40]. The document does not describe a methodology for the weighing process. However, it does mention that the weighing must be recorded by the reviewing authority.

## Domain: Outcome/benefit

We have combined the *outcome* and *benefit* domains into one domain because the relevant documents use "outcome" and "benefit" interchangeably without exception. The problem is that there are clear discrepancies related to the terminology of outcome and benefit and the respective expectations of one or the other. Separating outcome and benefit would have made our comparative overview more confusing. We shall, nonetheless, point out differences mentioned in the guidance documents wherever possible.

**Table 8. Swiss FSVO matrix for comparing strain (i.e., harm) against interests [39].**

| *Dignity of the animal is respected* | | Legitimate interests | | | |
|---|---|---|---|---|---|
| | | * | ** | *** | **** |
| **Strain** | * | No | Yes | Yes | Yes |
| | ** | No | No | Yes | Yes |
| | *** | No | No | No | Yes |

**Table 9.** *Outcome/benefit* domain in European HBA guidance documents.

| Domain | Outcome/Benefit | | EU | AT | BE | FR | NL | CH | UK |
|---|---|---|---|---|---|---|---|---|---|
| | | | *European countries* | | | | | | |
| *Domain* | **Outcome/Benefit** | **Q1** | • | • | • | • | • | • | • |
| | | **Q2** | – | – | – | – | – | • | • |

Q1: "Is the domain clarified/defined in the guidance document?".

Q2: "Does the guidance document provide a method for practically including the domain in HBA?". Answer options for Q1 and Q2 were either "•" (Yes) or "–" (No).

Art. 38 of the Directive requires that a project evaluation contains "an evaluation of the objectives of the project, the predicted scientific benefit or educational value"[1]. It also describes the HBA requirement as part of project evaluation and includes both outcome and benefit domains. HBA must assess "whether the harm to the animals [. . .] is justified by the expected outcome [. . .], and may ultimately benefit human beings, animals or the environment" [1]. The results of the outcome/benefit domain are presented in Table 9.

HBA-EU stipulates that benefits must be defined as either direct or indirect. There must be a description of the benefits, who the beneficiaries will be, how they will benefit, what the impact will be, and the expected timeline for attaining the benefits. The expected benefit must also be linked to the research purpose described in the Directive. HBA-EU also suggests that beneficiaries of the project must be listed in the application. There is, however, no answer to the question "*Who* will benefit from the work?" [16].

Like the harm dimension, outcome/benefit must be attributed a certain weight or value for it to be relevant to HBA's operationalization. HBA-EU recognizes that EU member states might allocate differing weights to different benefits during their review processes. These differences result from regional variances regarding the respective benefit's priorities. The guidance document does not go into further detail on how to practically attribute weight to benefits. It does, though, recommend that project evaluation processes require a case-by-case evaluation and describe the difficulty in weighing benefits: "Weighing of non-comparable, sometimes abstract benefits arising from different types of research programs is very difficult to perform objectively" [16].

HBA-AT asks applicants to assess the degree of benefit and to specify the beneficiaries of the research (humans, animals, and/or the environment). It also asks applicants to ensure that the research complies with and promotes the 3Rs. The project's overall purpose must also be addressed by designating it to a distinct area of research (e.g., basic research or translational research).

In HBA-BE, benefits are classified into five dimensions: (1) social, (2) socioeconomic, (3) scientific, (4) educational, or (5) safety and efficacy testing benefits. An assessment must also involve considering the likelihood of achieving the described benefits (vis-à-vis the scientific quality of the experiment). HBA-BE also states that the expected benefit to human beings, animals, or the environment must be taken into account during HBA [34]. However, HBA-BE does not contain a methodology for including the outcome/benefit domain in HBA.

HBA-FR also stipulates that a research proposal must identify relevant benefits. These must be analyzed in terms of questions like what exactly the benefits are, who the beneficiaries will be, the likelihood of achieving results, and when the results will be accessible and/or applicable. A method for applying this domain in the HBA cannot be found in the HBA-FR.

HBA-NL refers to benefits as "goals" [Authors' note: Translation DeepL]. Goals must be explained to justify the research purpose. Goals can be distinguished between primary, direct goals and ultimate, future goals. A direct goal is achieved at the end of the proposed project's

term, while an ultimate goal can be achieved sometime in the future once the project is completed [35].

HBA-NL likewise states that "the benefits of a project for humans, animals, and the environment must be included in the harm-benefit-analysis [. . .]" [35] [Authors' note: Translation DeepL]. That said, the document excludes a detailed methodology for practically implementing the outcome/benefit domain during HBA.

HBA-CH employs a different terminology; "legitimate interests of society" is used instead of the terms "benefit" and/or "outcome". The document describes legitimate interests of society as

> the preservation or protection of the life and health of human beings or animals, new knowledge concerning fundamental biological processes, and/or the protection of the natural environment [37].

A project proposal must establish that the expected gain of knowledge from the experiments is relevant to and can contribute to, at least, one of society's legitimate interests.

HBA-UK does not distinguish between outcome and benefit; the terms are used interchangeably. A specific benefit must be adequately described in the project proposal as a measurable outcome likely to result from the project. Benefits can be further divided into direct (project-related) benefits and indirect benefits. HBA-UK also describes a method for clarifying expected benefits. This involves asking certain detailed questions (Table 10).

HBA-UK acknowledges that "[w]eighting (determining the value) of particular benefits is usually more difficult than weighting of harms" [40]. It also acknowledges a subjective component:

> [T]he importance of work is subjective and changes with time and place, and depends on culture, environment, and the emergence of new knowledge and societal attitudes [40].

**Table 10. Overview of the questions described in HBA-UK to facilitate benefit assessments [40].**

| | |
|---|---|
| *What will the benefits of the work be?* | What data may be acquired? |
| | What drugs may be developed? |
| | Which scientific questions will be answered? |
| | What knowledge gaps might be filled? |
| | What is the project's output? |
| *Who and how many will benefit from the work?* | Other researchers? |
| | Human or veterinary patients? |
| | A relatively small set of patients (e.g., people with a rare genetic disease) or potentially millions of patients (e.g., a vaccine candidate for malaria)? |
| | The environment? |
| *How will the benefits accrue?* | Improved scientific knowledge or understanding? |
| | New or more efficacious therapies? |
| | Cheaper therapies? |
| | Impact on patients' quality of life? |
| *When will the benefits be achieved?* | A wide range of times (e.g., toxicological safety testing) or decades in the future (e.g., basic research potentially leading to future interventions). |
| *What benefits are not allowed?* | Developing or testing offensive weapons. |
| | Developing or testing alcohol or tobacco products. |
| | Testing cosmetics. |

HBA-UK also states that certain benefits can have a higher value than others (depending mostly on the degree of knowledge conferral).

## Domain: Ethical considerations

The Directive is clearly asking applicants to take ethical considerations into account during HBA. However, we could not find an interpretation of the term 'ethical considerations' in the legislation. We searched all guidance documents for a definition. We also analyzed the documents for recommendations on implementing ethical considerations during project evaluations (Table 11).

HBA-EU is not explicit about what exactly counts as ethical considerations. It also does not provide recommendations for implementation. The same applies to the Belgian and Swiss national guidance documents included in our study.

In HBA-AT, applicants are, however, asked to justify any harm inflicted. "[T]he ethical considerations behind it" should also be described [26] [Authors' note: Translation ours]. Applicants must also include a justification or rationale for the relevant benefit/outcomes. That said, what exactly ethical considerations entail and what their practical implementation involves remains unclear.

HBA-FR states that an "ethical evaluation" involves performing a comprehensive evaluation of project proposals while "taking into account ethical considerations in regard to the use of animals" [36] [Authors' note: Translation DeepL]. However, the document does not clarify what these ethical considerations mean for HBA, nor does it describe practical implementations.

The Dutch guide stipulates that reviewers must take ethical considerations into account when weighing harms and benefits. These considerations must also be explained. HBA-NL allows ethical considerations to be implemented during HBA from different perspectives and by applying various ethical theories [35]. The guidance document mentions some examples of ethical theories that could be used in deliberations and the decision-making process. These include consequentialism, deontology, and virtue ethics, e.g.: "A third perspective is virtue ethics. For persons thinking from this perspective, a person's character and the development of good character traits are central, virtues such as courage, patience, wisdom and honesty. This means that whether an action is good is determined not by the consequence of the action, but by the intentions and character of the person acting. This can come into play in terms of trusting that the applicant will make choices with integrity [...]" [35] [Authors' note: Translation DeepL]. We will argue later that this well-intended idea might actually lead to serious problems.

The UK document states as follows:

Section 5B(3)(d) of ASPA [Animal (Scientific Procedures) Act] requires that an HBA [...] is undertaken to assess whether the harm that would be caused to protected animals in

**Table 11. *Ethical considerations* in European HBA guidance documents.**

| Domain | Outcome/Benefit | | European countries | | | | | | |
|--------|-----------------|------|------|------|------|------|------|------|------|
| | | | EU | AT | BE | FR | NL | CH | UK |
| *Domain* | **Outcome/Benefit** | Q1 | – | – | – | – | • | – | – |
| | | Q2 | – | – | – | – | (•) | – | – |

Q1: "Is the domain clarified/defined in the guidance document?".

Q2: "Does the guidance document provide a method for practically including the domain in HBA?". Answer options for Q1 and Q2 were either "•" (Yes) or "–" (No). Only one answer was inconclusive; it is represented as "(•)".

terms of suffering, pain and distress is justified by the expected outcome taking into account ethical consideration and the expected benefit to human beings, animals or the environment [40].

However, no further detail for the domain of ethical considerations is provided.

## General overview and comparison of implementation status

All available guidance documents addressed our research questions for the harm domain (Table 12). There is consensus on the Directive's definition of 'harm' based on pain, suffering, and distress inflicted on animals. The Belgian, Dutch, French, and UK guidance documents have adapted this definition to include more detailed or nuanced description of harms by differentiating between direct and indirect harm and assigning a severity category. The Swiss document distinguishes between pathocentric harm and non-pathocentric harm. HBA-AT requires an exact allocation of animals to respective severity categories.

The HBA-EU requires a justification and provides an abstract example of a method. However, it does not provide further details; instead allowing for contextual variabilities. Austria's criteria catalog requires researchers to provide a rationale for harm (and benefit) but does not provide guidance on how to balance harm and outcome/benefit. The French guideline recommends a published method [10] for identifying harms and benefits. It also recommends displaying the relationship between harm and benefit in grid form. According to the Dutch guidance document, a justification is made if and when one can establish a relationship between harms and benefits. This is similar to the Swiss guidance document, which states that a justification has been made if the essentiality of the proposed experiments outweighs the anticipated harm inflicted on animals. A matrix helps to determine if a justification has been made. The Swiss document, therefore, clearly defines a methodology for this domain (cf. Table 8). HBA-UK defines the justificatory domain but does not describe a method for attaining it.

All guidance documents refer to either outcome or benefit, typically without discriminating between them. HBA-EU provides an overall recommendation on how to identify the benefits of a project. It also describes the difficulty involved in weighting them but leaves a clear methodology out. The Austrian, Belgian, and French guidelines follow the basic principles outlined

**Table 12. Results of the domain analysis for each country.**

| European countries | | Domains | | | | | | | |
|---|---|---|---|---|---|---|---|---|---|
| | | **Harm to animals** | | **Justification for harm** | | **Outcome/ Benefit** | | **Ethical considerations** | |
| | | **Q1** | **Q2** | **Q1** | **Q2** | **Q1** | **Q2** | **Q1** | **Q2** |
| | EU | • | • | • | (•) | • | – | – | – |
| | AT | • | • | • | – | • | – | – | – |
| | BE | • | • | – | – | • | – | – | – |
| | FR | • | • | • | – | • | – | – | – |
| | NL | • | • | • | – | • | – | • | (•) |
| | CH | • | • | • | • | • | • | – | – |
| | UK | • | • | • | – | • | • | – | – |

Q1: "Is the domain clarified/defined in the guidance document?".

Q2: "Does the guidance document provide a method for practically including the domain in HBA?". Answer options for Q1 and Q2 were either "•" (Yes) or "–" (No). Inconclusive answers are represented as "(•)".

in HBA-EU when it comes to describing benefits. The Dutch guidance document describes a method for placing outcome/benefit versus harm in a matrix illustrating values for all stakeholders affected by the experiment. The Swiss guideline focuses on gaining knowledge, which, in turn, contributes to society's interests. A value can be attributed to society's interests and placed in direct relation to harm inflicted on animals. The UK document generally aligns with HBA-EU. It provides details on assessing and defining benefits and mentions the difficulty involved in attributing a specific value to benefits. HBA-UK describes a detailed and differentiated overview of questions to identify their benefits and thereby facilitate their weighting (cf. Table 10).

The Dutch guidance document is the only one that describes the domain of ethical considerations in detail. The Austrian guidance for the domain of ethical considerations is rather vague. It is up to the applicant to decide which information will serve as underlying ethical considerations. Although stated in the Directive, the domain of ethical considerations is not further described in the European guidance document or mentioned in other documents.

As illustrated in Fig 8, all guidance documents address the HBA requirement's four domains similarly. There are, nonetheless, notable differences.

In sum, all guidelines have addressed our research questions related to the harm domain. The Austrian and French guidance documents are relatively similar to HBA-EU when it comes to addressing the four domains and the respective two research questions. That said, HBA-EU provides recommendations on methods for operationalizing the justificatory domain. This cannot be found in HBA-AT or HBA-FR (Fig 8). In fact, we found a straightforward process for operationalizing this domain only in HBA-CH, making it an outlier in this respect.

All guidance documents implement the outcome/benefit domain, but only HBA-CH and HBA-UK thoroughly address its practical application. HBA-BE addresses the harm and benefit domain's major components. That said, this document has the narrowest scope when compared with the other documents. HBA-NL is on the other end of the spectrum; it is the only document that attempts to address all four HBA requirement domains. However, it does not provide a transparent methodology for the outcome/benefit domain, the justification domain, and the ethical considerations domain. HBA-CH, in contrast, provides detailed answers for both research questions in three domains: the harm domain, the outcome/benefit domain, and the justification domain. However, the HBA-CH leaves out explicit details on implementing the ethical considerations domain. HBA-NL is an outlier when it comes to ethical considerations. It is the only document providing recommendations for practically applying a method in the ethical considerations domain. None of the other guidelines incorporated this domain. Only the Dutch, Swiss, and UK guidance documents go further than the European document when implementing at least one of the four domains. The Austrian, Belgian, and French documents align with most of the domains addressed in HBA-EU, but they do not go further than HBA-EU's recommendations.

## Discussion

Harmonizing legislation for research animals among EU member states has been a priority since the Directive's implementation. Documents accompanying the European legislation are supposed to provide guidance on issues such as HBA. Our study shows that the Directive is not being consistently implemented. Our results also suggest that harmonizing HBA across EU member states might not be possible presently.

While the majority of EU member states have transposed the HBA requirement in accordance with the Directive's stipulations, some countries are deviating from the wording.

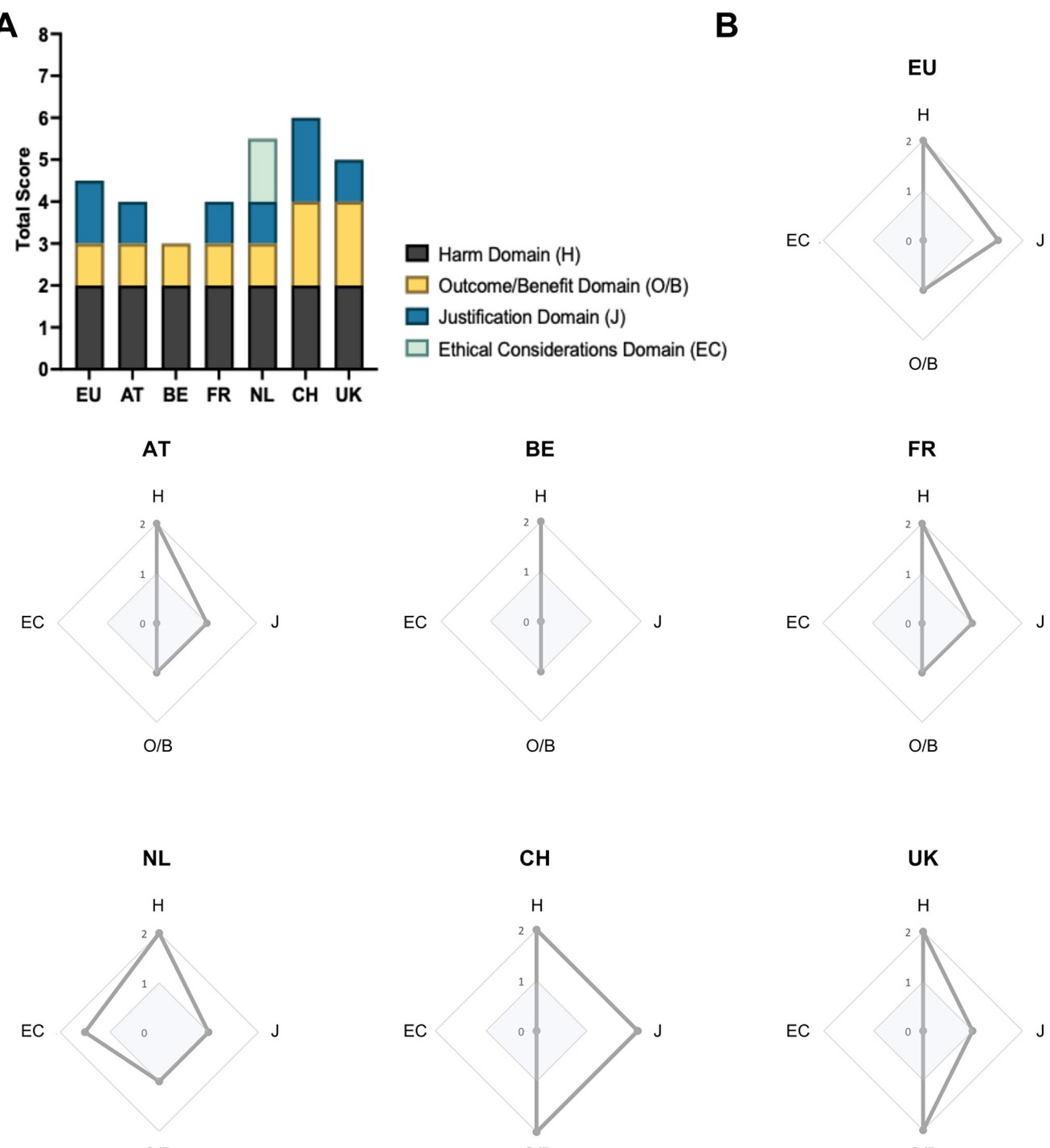

**Fig 8. Comparative domain analysis.** (A) Histogram: Transfer of answers to research questions Q1/Q2 for each domain into scores. Answer options for Q1/Q2 are "Yes" = 1 score (maximum score = 2) or "No" = 0 score. (B) Radar graphs: Scoring of research questions Q1/Q2 for *implementation* and *operationalization* illustrated individually for each guidance document. Answer options for Q1/Q2 are "Yes" = 1 score (maximum score = 2) or "No" = 0 score. The inner square (highlighted in light blue) represents the extent of domain *implementation*. The outer square demarcates a domain's *operationalization*. Q1: "Is the domain clarified/defined in the guidance document?"; Q2: "Does the guidance document provide a method for practically including the domain in HBA?". H = Harm domain, J = Justification domain, O/B = Outcome/Benefit domain, EC = Ethical Considerations domain.

The Finnish legislation ignores the outcome domain and approves a project if the benefit is in "an ethically justified proportion considering the harm to the animals" [29]. The principle of proportionality, which aims to ensure that animals are protected from measures that are disproportionate to the objectives, can be found throughout the Directive [45]. This principle is also applied in Finland's transposition of the HBA requirement.

The Danish legislation does not mention benefits or outcomes but rather speaks of the "usefulness" [30] [Authors' note: Translation DeepL] of a project. "Usefulness" might be interpreted as outcome, benefit, or both. The domain of ethical considerations is not part of the legislation's wording. It is, therefore, questionable whether the Directive's HBA requirement is being implemented in sufficient detail.

The Austrian legislation is an outlier. It aims to objectify HBA by imposing an additional legal requirement: a criteria catalog containing a standardized questionnaire for applicants. Nonetheless, it remains unclear whether the criteria catalog is useful to reviewing bodies and whether it consistently objectifies HBA.

One might interpret the above differences in some EU member states' national laws as a shortcoming of the Directive. The Directive's vague HBA requirement and its relatively unclear definition (or lack thereof) encourage some member states to employ their own interpretations of the HBA requirement. The fact that the Directive allows this leeway results in drastic, cross-national inconsistencies.

Our study has also highlighted apparent gaps regarding the practical implementation of HBA according to auxiliary guidelines. We can see that both the Directive and the overarching HBA-EU do not provide workable tools for HBA. Most European countries do not have any national or regional guidelines, and the guidance documents we could identify are far from comprehensive.

We have also seen that HBA-EU is insufficiently equipped to guide authorities through HBA. In fact, it is unsurprising that some national committees have created their own documents given the HBA-EU's scant and unclear guidance. As HBA-BE mentions, "the European Directive does not state, in any specific way, how to conduct an HBA and how to make sure that benefits will truly outweigh the harm [...]" [34]. It is a recipe for inconsistency if HBA-EU is not providing guidance on a practical level.

We have seen that there are differences between national and regional documents. This is because EU member states structure the review and authorization process differently [2, 3]. This could be causing inconsistent HBA implementations among EU member states. Diversifying the structure of reviewing levels into regional versus national competencies could also be affecting decision-making processes and disrupting HBA's harmonization. In fact, it is doubtable whether the HBA is carried out in a substantial way at all, as indicated in the Swedish case by Jörgensen et al. (2021) [19]: "We argue that the results show an unacceptably low level of compliance in the investigated applications with the legal requirement of performing both a harm–benefit analysis and applying the 3Rs within the decision-making process, and that by implication, public insight through transparency is not achieved in these cases."

Our analysis of the domains and our overview of the results (Table 12, Fig 8) indicates that only the harm domain is clearly defined and operationalized in countries where guidance documents are available. The other domains are all addressed differently, making it difficult to draw clear comparisons. It is unsurprising that the harm domain has been more extensively addressed. The debate around harm to animals and the associated legal ramifications has a long history [11, 42, 46, 47]. This is not the case with other HBA domains.

The Directive categorizes harm into four groups: non-recovery, mild, moderate, and severe [15]. One must, however, consider many additional factors to appropriately categorize harm. Required measures such as prospective and retrospective harm assessments can evaluate and

necessitate adjustments in severity categorizations. Defined categories and practically attributable values can also render the harm domain more understandable and relatable during HBA. Concrete examples can be found in the Directive's Annex VIII.

The justificatory procedure (or the weighing part) is the core of HBA. However, guidance documents (except in CH) are silent on a methodology. This supports the suspicion that HBA is carried out arbitrarily or not at all. This is not surprising given that the two main options for carrying out HBA–the discourse model and the metric model–have significant shortcomings [11]: (1) the discourse model is flexible and works without standardized methodology, but this means that HBA is applied inconsistently; (2) the metric model contains a strict methodology but lacks the flexibility often required to tailor debates toward issues that might not have been anticipated in the method [10, 11, 46, 48–50].

Most countries in the EU have installed committees that roughly follow the discourse model. As long as this seems to be working, they might be reluctant to implement a new methodology that potentially introduces new disadvantages. Nonetheless, some recent literature [11, 19] has given reasons to make the change. Despite topical discussions in the literature [7, 10, 11], we did not find any stipulations in the guidance documents related to the weighing process.

Only CH provides a systematic approach for the justificatory domain [37, 38]. It is, however, debatable whether this approach is sufficient. In any event, it does provide guidance to those carrying out HBA. Given that only one country implemented a tool to balance harms and benefits, it would be odd to assume that the various reviewing authorities followed the same principles during HBA.

From a philosophical point of view, the fact that harms and benefits cannot be weighed in one and the same currency is striking. There is ongoing debate about this so-called problem of *incommensurability* [7, 17, 51, 52]. One might wonder whether the idea of balancing harms and benefits is even appropriate for HBA. The following questions come to the fore when HBA is addressed in terms of weighing harms against benefits: What should we put on the scales? How do we select what goes on the scales? How much weight is attributed to the various aspects? What outweighs what? Are there limits (or stops) in the weighing process or can we place anything on the scales? Alternative approaches that replace the concept of 'scales' might be more successful. In fact, we find it difficult to imagine a less successful and more misguiding metaphor. In this regard, the debate might benefit from sustainability studies, where a lot of work has been devoted to developing accounts on balancing the three dimensions (i.e., social, economic, ecological) [53].

Things appear to be even vaguer in the outcome/benefit domain. As we have seen, there is a dearth of clear definitions and robust operationalization guidelines. This might be explained in at least three ways: (1) *Lack of routine*–the requirement that outcome/benefit be systematically considered when evaluating projects only dates to the Directive's transposition; (2) *Diversity*–benefits are more diverse than harms (which are more clearly identifiable and routinely assessable); (3) *Heterogeneity of legal purposes*–categories cannot be identified beforehand because projects' legal purposes are so heterogeneous.

A terminological problem emerges when we look at what Aurora Brønstad et al. call the "promise dimension" of projects, namely outcome and benefit [7]. Despite initial appearances, outcome and benefit are not particularly similar concepts [54, 55]. A project's outcome typically consists of data and new insights, that is, *knowledge*. These results are not synonymous with benefits [17, 54, 55]. In the HBA's formulation, benefits relate to "making a positive difference" for humans, animals, or the environment.

In conclusion, HBA's formulation leaves us with the puzzle of whether and how outcome and benefit might be differentiated and weighted against each other [56, 57]. Most of the

guiding documents do not address differences between the two concepts. This leaves applicants with the open question of whether results from *basic* research (i.e., knowledge gain) are more or less weighty than results from *applied* research (i.e., benefits). In fact, basic research is much more likely to achieve a promised outcome (knowledge) because negative results can add to our body of scientific knowledge. The relationship between scientific outcome and practical benefit is much vaguer in applied research (or translational research), and its realization is also contingent [54]. Basic research sets realistic and achievable goals, but this might not always be the case with applied research. As such, we advise that leading guidelines should incorporate a clear definition of these two distinct terms. Doing so can avoid further confusion during project applications and review processes.

A striking finding is that most guidance documents do not address the ethical considerations domain (only HBA-NL clearly defines and attempts a methodology). However, implicitly, HBA tends to exhibit the following ethical features:

1. Consequentialism. The net of positive and negative consequences defines the quality of an experiment.

2. Pathocentrism. The Directive predominantly focuses on animals' negative subjective experiences.

3. Hierarchy. Not all animals are placed on the scale, and as stated in Art. 1 (3), the Directive shall apply to "[. . .] (a) live non-human vertebrates, including (i) independently feeding larval forms; and (ii) foetal forms of mammals as from the last third of their normal development; (b) live cephalopods" [1]. Hence, the Directive does not consider humans and some non-human animals to be 'animals' in the relevant sense, e.g., insects or mollusks are not included [58].

HBA-NL suggests including reflections based on ethical theories such as virtue ethics, consequentialism, or deontology. However, this well-intended proposal comes along with a salient risk. If parties consider ethical considerations that go beyond what is legally required, then decisions might be made without a legal basis. While ethical principles may be considered and can overlap with legal norms, they are not identical. Authorities must prioritize legal concerns or risk violating the rule of law.

Policy documents are generally not part of scientific publications. Our literature survey–with defined search terms and streams–has provided little help in identifying them. It is possible that other national or regional guidance documents are being used, ones that were not included in our study. In addition, some national documents were not available in English. Translation software proved somewhat unreliable in certain cases, leading to potential translation errors. This may be the case in both the guidance documents and the relevant national legislations.

We can conclude that HBA is really guided in only a few European countries. Yet, this guidance remains unclear about most domains and the key factors that can make HBA work. If an international research community wants to adopt a consistent approach, then they will have to debate the correct methodology. We have outlined the difficulties involved in agreeing on a consistent strategy.

Introducing HBA as a legal necessity may contribute to inconsistencies across the board. Authorities are largely in the dark; they do not have concise, definitive, and actionable guidance. A pivotal change would be to amend the HBA requirement, which could render HBA a relatable and useful tool during the review process. Overarching guidelines should not recommend methodologies based on an assumption of practical applicability. Methodologies should rather be based on scientific results that have proven those methodologies to be successful

during decision-making processes. One possibility is to retrospectively assess the model as previously carried out on the existing UK version [59]. Both proposed HBA models–the discourse model and the metric model–have evident advantages and disadvantages. One or the other might not be the proper solution. A combination of both could facilitate a consistent review process by (1) allowing more flexibility during deliberations and (2) transparent and clear dialogue [11]. This model and its efficacy (vis-à-vis safeguarding against unnecessary harm inflicted on animals and the 3Rs) is yet to be determined and should therefore be part of forthcoming studies.

There is no doubt a need for practically efficacious HBA. This remains to be accomplished. The question is whether this is mission impossible.

## Supporting information

**S1 File.**
(PDF)

**S2 File.**
(PDF)

**S3 File.**
(PDF)

**S4 File.**
(PDF)

**S5 File.**
(PDF)

**S6 File.**
(PDF)

**S7 File.**
(PDF)

## Acknowledgments

We would like to thank Helena Röcklinsberg and Svea Jörgensen for critical comments on the preliminary data presented at the Veterinary Ethics Conference in September 2023 in Vienna, Austria. Further, we would like to thank Camilla Predella for her valuable input on the data visualization.

## Author Contributions

**Conceptualization:** Dominik Hajosi, Herwig Grimm.

**Data curation:** Dominik Hajosi.

**Formal analysis:** Dominik Hajosi.

**Investigation:** Dominik Hajosi.

**Methodology:** Dominik Hajosi, Herwig Grimm.

**Supervision:** Herwig Grimm.

**Writing – original draft:** Dominik Hajosi, Herwig Grimm.

**Writing – review & editing:** Dominik Hajosi.

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
