## [Decision Letter · Decision Letter 0]

6 Dec 2023

PONE-D-23-34073Mission impossible accomplished? A European cross-national comparative study on the integration of the harm-benefit analysis into law and policy documentsPLOS ONE

Dear Dr. Hajosi,

Thank you for submitting your manuscript to PLOS ONE. After careful consideration, we feel that it has merit but does not fully meet PLOS ONE’s publication criteria as it currently stands. Therefore, we invite you to submit a revised version of the manuscript that addresses the points raised during the review process.

We look forward to receiving your revised manuscript.

Kind regards,

Jianhong Zhou

Staff Editor

PLOS ONE

Journal Requirements:

Reviewers' comments:

Reviewer's Responses to Questions

**Comments to the Author**

1. Is the manuscript technically sound, and do the data support the conclusions?

Reviewer #1: Yes

Reviewer #2: Yes

2. Has the statistical analysis been performed appropriately and rigorously? 

Reviewer #1: N/A

Reviewer #2: N/A

3. Have the authors made all data underlying the findings in their manuscript fully available?

Reviewer #1: Yes

Reviewer #2: Yes

4. Is the manuscript presented in an intelligible fashion and written in standard English?

Reviewer #1: No

Reviewer #2: Yes

5. Review Comments to the Author

Reviewer #1: This is an excellent piece of work which will be very helpful to international agencies wanting to adopt some form of approach to justification of animal-based projects. It highlights the importance of definitions of terminology, e.g. outcome, benefit, basic research, applied research. It also raises the challenges inherent in attempting to "weigh" harms and benefits, pointing to the "incommensurability" of harms and benefits. The low uptake of a true HBA analysis by EU nations speaks to these challenges in attempting to weigh "apples and oranges". While not offering solutions per se, this manuscript points to the need to develop methodology that guides acceptable evaluation of potential project; likely a mix of what the authors here call "discourse models" and "metric models".

The manuscript is very well written and clear. Please just check line 697, the sentence as written doesn't quite make sense to me.

Reviewer #2: The paper is interesting and devoted to an essential difficulty in the practice of ethical review under the EU Directive 63/2010. That is why it makes a valuable contribution to the ongoing discussion and the potential improvement of the legal framework of animal experimentation in the EU. My main comments concern the discussion part, which partially exceeds what can be inferred from the data gathered and analyzed. Namely, I find the lines 832-854 highly questionable. The ethical consideration must be taken into account within the HBA and the local legal context, and even reference to various ethical theories may be a fully legitimate part of such analysis as long as they justify conclusions that remain within the discretion left by the law in question. I'd suggest skipping the above lines, which does not affect the main reasoning or value added by the paper.

On the other hand, the central claims in favor of making HBA more operative are justified and well-founded. I don't see any grounds, however, either in the paper or elsewhere, for the claim (line 865) that no uniform solution is achievable (or should be sought). It should be either justified or removed. Furthermore, an interesting and reasonable suggestion that discursive and metric models might be combined could be somehow elucidated rather than only mentioned. That seems a legitimate expectation from the paper examining the gaps and inconsistencies of the existing EU framework and successfully spotting its main weaknesses.

6. PLOS authors have the option to publish the peer review history of their article (what does this mean?). If published, this will include your full peer review and any attached files.

Reviewer #1: **Yes: **Gilly Griffin

Reviewer #2: No

---

## [Author Response · Author response to Decision Letter 0]

13 Dec 2023

Response to Reviewers

The authors would like to sincerely thank the Editor and Reviewers for their thoughtful consideration of our manuscript and the helpful feedback that they have provided. The comments will benefit the quality of this manuscript. Below, you will find our responses to all comments made by the Editor and Reviewers and notes on the changes made in the revised version of the manuscript.

Editor comments (comments made by the editor are shown in italics, and responses by the authors are shown in blue):

Our response: Thank you for the comment and suggestion. Reviewing the recommended templates, we have ensured that our manuscript meets PLOS ONE’s style requirements, including file naming.

Our response: Thank you for the comment and suggestion. Due to the changes suggested by the Reviewers, we have deleted eight references. Additionally, we have updated the year of publication for one reference (line 1101 in the revised manuscript). We have thoroughly double-checked our reference list for accuracy and completeness. We can confirm that neither of the papers cited have been retracted.

Reviewer comments (comments made by reviewers are shown in italics, and responses by the authors are shown in blue):

Reviewer #1

1. This is an excellent piece of work which will be very helpful to international agencies wanting to adopt some form of approach to justification of animal-based projects. It highlights the importance of definitions of terminology, e.g. outcome, benefit, basic research, applied research. It also raises the challenges inherent in attempting to "weigh" harms and benefits, pointing to the "incommensurability" of harms and benefits. The low uptake of a true HBA analysis by EU nations speaks to these challenges in attempting to weigh "apples and oranges".

Our response: The authors thank the Reviewer for the very favorable comments.

2. While not offering solutions per se, this manuscript points to the need to develop methodology that guides acceptable evaluation of potential project; likely a mix of what the authors here call "discourse models" and "metric models".

Our response: We thank the Reviewer for this comment. We agree that there is a need to develop a methodology for the HBA, and a combination of the discourse and metric model may be a new path moving forward.

3. The manuscript is very well written and clear. Please just check line 697, the sentence as written doesn't quite make sense to me.

Our response: We thank the Reviewer for inviting this clarification. We have changed this sentence (lines 697-698 in the revised manuscript) to ensure a better understanding.

Reviewer #2

1. The paper is interesting and devoted to an essential difficulty in the practice of ethical review under the EU Directive 63/2010. That is why it makes a valuable contribution to the ongoing discussion and the potential improvement of the legal framework of animal experimentation in the EU.

Our response: The authors thank the Reviewer for this very positive comment and for outlining the relevance of this manuscript.

2. My main comments concern the discussion part, which partially exceeds what can be inferred from the data gathered and analyzed. Namely, I find the lines 832-854 highly questionable. The ethical consideration must be taken into account within the HBA and the local legal context, and even reference to various ethical theories may be a fully legitimate part of such analysis as long as they justify conclusions that remain within the discretion left by the law in question. I'd suggest skipping the above lines, which does not affect the main reasoning or value added by the paper.

Our response: The authors thank the Reviewer for this insightful comment and the suggestion, which we took into consideration. We deleted the questionable lines 836-854, as suggested by the Reviewer, and only kept the main point that ethical consideration must remain within the legal framework (an aspect that the Reviewer and authors agree on). Utilizing ethical accounts as part of the analysis for the decision-making is a positive and legitimate thing as long as it remains within the legal framework. Therefore, the authors decided to delete 836-854, which does not affect the main reasoning of the paper. We reworded lines 832-836 to keep the mentioned point (lines 834-839 in the revised manuscript).

3. On the other hand, the central claims in favor of making HBA more operative are justified and well-founded.

Our response: We thank the Reviewer for this comment in support of our arguments.

4. I don't see any grounds, however, either in the paper or elsewhere, for the claim (line 865) that no uniform solution is achievable (or should be sought). It should be either justified or removed.

Our response: We thank the Reviewer for this comment and agree with the assessment. As suggested by the Reviewer, we removed line 865 from our manuscript.

5. Furthermore, an interesting and reasonable suggestion that discursive and metric models might be combined could be somehow elucidated rather than only mentioned. That seems a legitimate expectation from the paper examining the gaps and inconsistencies of the existing EU framework and successfully spotting its main weaknesses.

Our response: The authors thank the Reviewer for this comment. The cited paper by Grimm et al. (2018; doi.org/10.1177/00236772187830) proposes the combination of the discourse and metric model. This might represent a novel methodology for the HBA. Accordingly, the point raised by the Reviewer has been addressed in part by Grimm et al. (2018). Although such a detailed methodology would be highly appreciated by many, developing it would go beyond the scope of this paper. Based on the Reviewer’s comment, we have included the notion for forthcoming studies in the Discussion (line 907 in the revised manuscript).

---

## [Decision Letter · Decision Letter 1]

4 Jan 2024

Mission impossible accomplished? A European cross-national comparative study on the integration of the harm-benefit analysis into law and policy documents

PONE-D-23-34073R1

Dear authors,

I am pleased to inform you that both reviewers enjoyed the manuscript very much and endorsed the revised manuscript for publication.

Thank you for choosing Plos ONE journal to publish your study.

Best regards,

António Machado

Reviewers' comments:

Reviewer's Responses to Questions

**Comments to the Author**

1. If the authors have adequately addressed your comments raised in a previous round of review and you feel that this manuscript is now acceptable for publication, you may indicate that here to bypass the “Comments to the Author” section, enter your conflict of interest statement in the “Confidential to Editor” section, and submit your "Accept" recommendation.

Reviewer #1: All comments have been addressed

Reviewer #2: All comments have been addressed

2. Is the manuscript technically sound, and do the data support the conclusions?

Reviewer #1: Yes

Reviewer #2: Yes

3. Has the statistical analysis been performed appropriately and rigorously? 

Reviewer #1: N/A

Reviewer #2: N/A

4. Have the authors made all data underlying the findings in their manuscript fully available?

Reviewer #1: Yes

Reviewer #2: Yes

5. Is the manuscript presented in an intelligible fashion and written in standard English?

Reviewer #1: Yes

Reviewer #2: Yes

6. Review Comments to the Author

Reviewer #1: I am satisfied that the authors fully understood the reviewers comments, and addressed them satisfactorily

Reviewer #2: I am fully satisfied with the authors' responses and revisions. I find the manuscript ready for publication.

7. PLOS authors have the option to publish the peer review history of their article (what does this mean?). If published, this will include your full peer review and any attached files.

Reviewer #1: **Yes: **Gilly Griffin

Reviewer #2: **Yes: **Tomasz Pietrzykowski

---

## [Editor Report · Acceptance letter]

31 Jan 2024

PONE-D-23-34073R1 

PLOS ONE

Dear Dr. Hajosi, 

I'm pleased to inform you that your manuscript has been deemed suitable for publication in PLOS ONE. Congratulations! Your manuscript is now being handed over to our production team.

Kind regards, 

on behalf of

Dr. António Machado 

Academic Editor

PLOS ONE